# High *ECM2* Expression Predicts Poor Clinical Outcome and Promotes the Proliferation, Migration, and Invasiveness of Glioma

**DOI:** 10.3390/brainsci13060851

**Published:** 2023-05-24

**Authors:** Junsheng Li, Siyu Wang, Qiheng He, Fa Lin, Chuming Tao, Yaowei Ding, Jia Wang, Jizong Zhao, Wen Wang

**Affiliations:** 1Department of Neurosurgery, Beijing Tiantan Hospital, Capital Medical University, Beijing 100070, China; 2China National Clinical Research Center for Neurological Diseases, Beijing 100070, China; 3Center of Stroke, Beijing Institute for Brain Disorders, Beijing 100070, China; 4Beijing Key Laboratory of Translational Medicine for Cerebrovascular Disease, Beijing 100070, China; 5Beijing Translational Engineering Center for 3D Printer in Clinical Neuroscience, Beijing 100070, China; 6Department of Neurosurgery, Second Affiliated Hospital of Soochow University, Suzhou 215004, China; 7Department of Clinical Diagnosis, Laboratory of Beijing Tiantan Hospital, Capital Medical University, Beijing 100070, China; 8Savaid Medical School, University of the Chinese Academy of Sciences, Beijing 101408, China

**Keywords:** *ECM2*, lower-grade glioma, extracellular matrix, prognosis, biomarker

## Abstract

Objective: Glioma is the most prevalent and fatal intracranial malignant tumor. Extracellular matrix protein 2 (ECM2) has rarely been studied in gliomas. Therefore, we explored the role of ECM2 in lower-grade gliomas (LGGs). Methods: The RNA-seq and clinicopathology data were obtained from the TCGA database. The immunohistochemical (IHC) staining was used to verify the expression of ECM2. Functional enrichment analyses, immune-related analyses, drug sensitivity, and mutation profile analyses were further conducted. Cox regression and Kaplan–Meier curves were utilized for survival analyses, while four external datasets were used to validate the prognostic role of ECM2. Furthermore, qRT-PCR, CCK-8, wound healing, and transwell assays were performed to confirm the function of ECM2 in gliomas. Results: The study found a significant upregulation of ECM2 expression with increasing glioma grades and a significant association between ECM2 expression and tumor immune infiltration. Cox regression verified the prognostic role of ECM2 in LGG patients (HR = 1.656, 95%CI = 1.055–2.600, *p* = 0.028). High ECM2 expression was significantly associated with poor outcome (*p* < 0.001). Four external datasets validated its prognostic value. After the knockdown of ECM2, the functional experiments showed a significant decrease in proliferation, migration, and invasion in glioma cell lines. Conclusion: The study suggested the potential of ECM2 as a novel immune-associated prognostic biomarker and therapeutic target for glioma patients.

## 1. Introduction

Glioma is a highly aggressive neoplastic disease originating from glial cells and is considered as the most common type of brain tumor [1]. According to the World Health Organization (WHO) classification, grade 2 and 3 gliomas are referred to as lower-grade gliomas (LGG), while grade 4 is known as glioblastoma (GBM) [2]. Although LGG shows a more indolent behavior and slower growth than GBM, the local recurrence and malignant transformation seem to be inevitable due to the infiltrating growth pattern [3]. As a result of glioma heterogeneity, patients have different sensitivities to chemotherapy and radiotherapy. Despite the emergence of new treatment options, such as targeted therapy and immunotherapy, the therapeutic efficacy remains limited and the overall survival (OS) of patients varies widely [4,5,6]. Thus, there has been an urgent need to identify novel biomarkers for prognostic prediction and individualized treatment in glioma patients.

Increasing evidence has shown that the extracellular matrix (ECM) plays an important role in the pathogenesis, development, and therapy of tumors. Extracellular matrix protein 2 (ECM2) is located on chromosomal 9q22.3 [7]. As an extracellular matrix component, ECM2 is involved in the regulation of cell proliferation and differentiation [8]. Recent studies have found an association between ECM2 and colon adenocarcinoma, hepatocellular carcinoma, and cervical cancer [9,10,11]. Additionally, ECM2 has been considered to be associated with idiopathic pulmonary fibrosis and pulmonary artery hypertension [12,13]. A previous study reveals the association between ECM2 and lymphopoiesis that ECM2 enhances the proliferation of mature B cells [14]. Therefore, we consider that ECM2 plays a potential role in the tumor immune microenvironment and affects the prognoses in LGG patients. However, there are few studies focused on the role of ECM2 in gliomas and its prognostic value for LGG patients is still unclear.

In this study, we investigate the role of ECM2 in gliomas and its prognostic value for LGG patients. We perform functional enrichment analyses, immune analyses, and drug sensitivity of ECM2 to identify its potential function. We further identify and verify its prognostic value using four external datasets. The function of ECM2 in glioma is confirmed with in vitro experiments. Overall, our study suggests that ECM2 is a promising prognostic indicator and potential therapeutic target for glioma patients.

## 2. Materials and Methods

### 2.1. Gene Expression Analysis in Gliomas

The GEPIA platform (http://gepia.cancer-pku.cn/index.html, accessed on 20 November 2022), an interactive platform for analyzing RNA-sequencing expression data of tumors and normal samples from the TCGA and GTEx projects, was used to compare the ECM2 expression levels of LGG and GBM samples with that of normal tissues. The level 3 HTSeq-count data and corresponding clinical features of 529 LGG samples were obtained from the TCGA database. Further analyses were conducted to compare the expression levels between different grades of glioma. The protein expression levels of ECM2 between different grades of glioma were confirmed by immunohistochemical (IHC) staining from the Human Protein Atlas (HPA) database (http://www.proteinatlas.org/, accessed on 20 November 2022). We performed functional enrichment analyses, immune-related analyses, mutant profiles, and drug sensitivity analyses based on the TCGA database.

### 2.2. Differentially Expressed Gene (DEG), Genetic Interaction, and Functional Enrichment Analyses

All the samples were divided into low and high ECM2 expression groups based on the median expression level of ECM2. The DESeq2 package was used to determine DEGs between the two groups with |logFC| > 1 and adjusted *p*-value < 0.05 [15]. GO and KEGG enrichment analyses were conducted by the ClusteProfiler package to investigate the potential biological functions [16,17]. The GeneMANIA database has been used to investigate genetic interaction, protein–DNA interaction, and protein–protein interaction. We explored the gene interaction network and aggregate function-similar genes with ECM2 by the GeneMANIA platform. Gene-set enrichment analyses (GSEA) were performed to explore the potential functional annotations and signaling pathways by ClusteProfiler [18]. The repetition of gene-set permutation was set to 1000 times in each analysis. The filter conditions were defined as adjusted *p*-value < 0.05 and FDR value < 0.1.

### 2.3. Immune-Related Analyses

The immune infiltration analyses were performed by single-sample gene-set enrichment analysis (ssGSEA) with GSVA package [19]. The levels of 24 types of infiltrating immune cells were compared between low and high ECM2 expression groups [20]. The correlation between ECM2 expression and immune cells was also analyzed using Spearman correlation tests. The immune infiltration analyses were validated by the TIMER database, a comprehensive resource for systematic immune infiltration estimation with 10,897 samples across 32 cancer types from TCGA [21]. Furthermore, Kaplan–Meier plots were performed to evaluate the combined effect of ECM2 expression level and immune infiltration on survival probabilities. The correlations between the expression of ECM2 and documented immune checkpoints were also analyzed using Spearman correlation tests [22], and the expression of immune checkpoints was displayed using a heat map by the ggplot2 package.

### 2.4. Drug Sensitivity Analyses

Genomics of Drug Sensitivity in Cancer (GDSC) was used to compare the sensitivity of patients to different chemotherapeutic agents between low and high ECM2 expression groups, which was assessed with half-maximal inhibitory concentration (IC50) by the pRRophetic package [23].

### 2.5. Mutation Profiles in Different ECM2 Expression Levels

The mutation profiles of LGG samples from the TCGA database were shown. The top-10 types of mutant genes were selected in the low and high ECM2 expression, and the frequencies of the top-10 mutant genes in LGGs were compared between these two groups.

### 2.6. Development of Prognostic Model

The prognostic role of ECM2 was evaluated by Cox regression and Kaplan–Meier analyses. Then, the same clinical characteristics were further included into the nomogram generated by the RMS package and survival package to predict the OS rates at 1 year, 2 years, and 3 years [24]. The calibration plot evaluated the accuracy by mapping the predictive probabilities to observed rates. In the calibration plot, the bootstrap method was set for 1000 times. Time-dependent receiver operating characteristic (ROC) curves were used to assess the nomogram predictive accuracy by the timeROC package.

### 2.7. Survival Analysis Validation

Four external datasets were used to validate the prognostic value of *ECM2* in LGGs, including 443 samples from the CGGA mRNAseq-693 dataset, 182 samples from the CGGA mRNAseq-325 dataset, 162 samples from the REMBRANDT cohort, and 107 samples from the GSE16011 dataset. Cox regression and Kaplan–Meier curves were used for survival analyses.

### 2.8. Cell Culture and Transfection

U87 and U251 cell lines were obtained from the American Type Culture Collection (ATCC, Manassas, VA, USA) and cultured in DMEM (gibco, Grand Island, NE, USA) supplemented with 10% FBS (gibco, Grand Island, NE, USA), penicillin, and streptomycin. The cells were placed in an incubator at 37 °C and 5% carbon dioxide and allowed to grow to confluence before transfection. For the siRNA knockdown experiment, the U87 or U251 cell lines were transfected with siRNA against ECM2 (Syngentech, Beijing, China) using Lipofectamine 3000 (Invitrogen, Carlsbad, CA, USA). The control groups were transfected with siNC (negative control). Transfection efficiency was assessed by quantitative real-time PCR (qRT-PCR).

### 2.9. qRT-PCR

Total RNA was isolated using TRIzol reagent (Invitrogen, Waltham, MA, USA) at 48 h after siRNA transfection, followed by cleaning with gDNA Eraser (Takara, Kyoto, Japan). The purified RNA was reverse transcribed by PrimeScript™ RT reagent Kit (Takara, Kyoto, Japan) after measuring the RNA concentration. The qRT-PCR was performed by TB Green Premix Ex Taq (Takara, Kyoto, Japan) with specific primers and the QuantStudio™ 3 System (Applied Biosystems, Waltham, MA, USA). GAPDH was used as a reference gene. All amplification reactions were carried out over 40 cycles (a hold stage of 30 s at 95 °C, then a two-step program of 3 s at 95 °C, 34 s at 60 °C). The mRNA expression for transcripts was calculated by the ΔΔCt method. The primer sequences were: GAPDH Forward AATGACCCCTTCATTGAC; GAPDH Reverse TCCACGACGTACTCAGCGC; ECM2 Forward GTTCCGAATGCCCTCTCGAT; ECM2 Reverse TCAGCGGTGGTATCTGGGTA.

### 2.10. Cell Counting Kit-8 Assay

The CCK-8 (Applygen, Beijing, China) assay was applied to determine cell proliferation. The microplate reader Synergy™ was used to measure the 450 nm absorbance after incubation for 24 h, 48 h, and 72 h.

### 2.11. Wound Healing Assay

The wound healing assays were performed with Ibidi Culture-Insert (Ibidi, Gräfelfing, Germany). The U87 or U251 cell lines were suspended in a complete medium at 300,000 cells/mL and 70 μL cell suspensions were pipetted into each chamber of the cell culture insert. After 12 h, the Culture-Insert was gently removed using sterile tweezers. Then, the well was filled with a serum-free medium to exclude the effect of cell proliferation. Images were photographed at 12 h after the scratch was made using an inverted-phase microscope (IX51, OLYMPUS, Japan), and the percentage of the reduced area was measured using NIH ImageJ software (version 1.52a).

### 2.12. Transwell Assay

The cell migration and invasion abilities were determined using a transwell chamber (8 μm, 24-well insert, Costar, Washington, DC, USA). For the invasion assay, matrigel (BD, Becton, Dickinson and Company, Franklin Lakes, NJ, USA) was diluted with serum-free medium (1:40), mixed, and used to coat the insert-chamber membrane. Then, cells after 48 h transfection were added to the upper chamber, and a medium containing 10% FBS was added to the lower. For the migration assay, cells were incubated for 48 h in a cell culture incubator with 4% formaldehydum polymerisatum. 0.1% crystal violet was used for the fixation and staining of the invading or migrating cells.

### 2.13. Statistical Analyses

All of the statistical analyses in this study were carried out by the R project. The expression of *ECM2* between different groups has been performed by Wilcoxon rank-sum tests and Kruskal–Wallis tests. Spearman correlation tests were used to assess the correlation. Pearson chi-square tests were used for the comparison of baseline clinical characteristics between the two groups. The prognostic role of *ECM2* was confirmed by hazard ratios (HRs) and 95% confidence intervals (CIs) of Cox regression. The survival distributions were estimated by Kaplan–Meier analyses with log-rank tests. A two-sided *p* value < 0.05 was considered to be statistically significant for analyses in this study.

## 3. Results

### 3.1. ECM2 Expression Levels of Gliomas

The expression of ECM2 was found to be significantly upregulated in both LGG and GBM samples compared to normal tissues (Figure 1a). Further analyses revealed that ECM2 expression was significantly higher in GBM than in LGG (*p* < 0.001, Figure 1b). Subsequently, we analyzed the ECM2 expression levels in WHO grades 2, 3, and 4, and found that ECM2 was significantly upregulated with an increasing grade (***p*** < 0.001 for all). The expression levels of ECM2 in glioma were verified by IHC analyses in the HPA database. It confirmed that ECM2 expression increased with the glioma grade (Figure 1c).

### 3.2. Identification of DEGs and Functional Enrichment Analyses

According to the screening criteria, we identified 3944 DEGs between the two different ECM2 expression groups, including 2466 upregulated and 1478 downregulated genes (Figure 2a). The DEGs were further included for the GO and KEGG analyses. The results of the GO analyses showed an association with immune function (Figure 2b). The results of the KEGG analyses were shown (Figure 2c). The genetic networks performed by GeneMANIA showed that FGF7 had the most complex connection with ECM2 (Figure 2d). Additionally, the hallmark items in GSEA analyses showed enrichment in immune functions and signaling (Figure 2e).

### 3.3. Immune-Related Analyses

The results of immune infiltration by ssGSEA showed that the infiltrating levels of most immune cells were commonly higher in the high *ECM2* expression group (Figure 3a). Further analyses showed that macrophages, eosinophils, neutrophils, aDCs, iDCs, Th17 cells, cytotoxic cells, T helper cells, NK cells, T cells, Tgd, NK CD56dim cells, B cells, Tem, Th2 cells, and Tcm were positively correlated with *ECM2* expression, whereas pDCs, NK CD56bright cells, and TReg were negatively correlated (Figure 3b).

The results of immune cell infiltration in the TIMER database showed that infiltrating immune cells were significantly positively associated with ECM2 expression (Figure 3c). We further analyzed the combined effect of ECM2 expression and immune infiltration on OS rates (Figure 3d). Compared to low immune infiltration, high neutrophil infiltration was significantly associated with poor prognoses in both low and high ECM2 expression groups (*p* < 0.05 for both). In the high ECM2 expression group, high macrophage and dendritic cell infiltration led to worse outcomes, and the high CD4 T cell infiltration led to a worse prognosis in the low ECM2 expression group (*p* < 0.05 for all). However, there was no significant difference in survival probabilities with B cell infiltration and CD8 T cell infiltration in both low and high ECM2 expression groups (*p* > 0.05 for all).

Furthermore, we analyzed the association between ECM2 expression and immune checkpoints (Figure 3e). The results showed significant positive correlations between ECM2 and most immune checkpoints. The heat map showed significant enhancement of most immune checkpoints in the high ECM2 expression group.

### 3.4. Drug Sensitivity Analyses

The results revealed that the IC50 values of Bortezomib, Camptothecin, Cisplatin, Cyclopamine, Etoposide, Gemcitabine, Paclitaxel, Rapamycin, Roscovitine, and Temozolomide (TMZ) were significantly lower in the high ECM2 expression groups (*p* < 0.001 for all, Figure 4), indicating that patients with high ECM2 were more sensitive to chemotherapy.

### 3.5. Mutation Profiles According to ECM2 Expression

To investigate the mutation profiles in the low and high ECM2 expression groups, we identified the top-10 mutant genes in each group (Figure 5a,b). In the low ECM2 expression group, the top-10 mutant genes were IDH1 (89.5%), CIC (36.8%), TP53 (32.8%), ATRX (24.3%), FUBP1 (16.6%), NOTCH1 (12.1%), TTN (10.1%), IDH2 (7.7%), PIK3CA (7.7%), and MUC16 (6.9%). In contrast, the top-10 mutant genes in the high ECM2 expression group were IDH1 (71.9%), TP53 (65.1%), ATRX (46.8%), TTN (15.7%), EGFR (11.9%), PTEN (8.5%), NF1 (8.1%), PIK3CA (8.1%), MUC16 (6.8%), and FLG (6.4%).

We further compared the frequencies of the top-10 mutant genes in LGGs between these two groups (Figure 5c). The results showed that the mutation frequencies of IDH1, CIC, FUBP1, NOTCH1, ARID1A, and IDH2 were significantly higher in the low ECM2 expression group (*p* < 0.05 for all). Conversely, the mutation frequencies of TP53, ATRX, EGFR, and PTEN were significantly higher in the high ECM2 expression group (*p* < 0.05 for all).

### 3.6. Association between Clinical Features and ECM2 Expression

We analyzed the clinical characteristics of LGGs between low and high ECM2 expression groups in the TCGA database (Table 1). The results showed that the rates of WHO grade 3, IDH wild-type, and 1p19q non-codeletion were significantly higher in the high ECM2 expression group (*p* < 0.001 for all).

Furthermore, we analyzed the expression levels of ECM2 in different clinical groups (Figure 6). The results showed that the expression levels of ECM2 were significantly upregulated in WHO grade 3, IDH wild-type, and 1p19q non-codeletion groups (*p* < 0.001 for all). However, no significant difference in the ECM2 expression was observed in different age levels and genders (*p* > 0.05 for both).

### 3.7. Prognostic Role of ECM2 and Nomogram Development

Univariate and multivariate Cox analyses identified that ECM2 was independently associated with poor OS in LGG patients (HR = 1.656, 95%CI = 1.055–2.600, *p* = 0.028, Table 2). Kaplan–Meier curves also confirmed that high ECM2 expression was significantly correlated with a worse prognosis compared to low ECM2 expression (*p* < 0.001, Figure 7a). We further analyzed the survival distribution between low and high ECM2 expression in different radiotherapy and chemotherapy subgroups (Figure A1). The results showed that a significant correlation between high ECM2 expression and poor prognosis in the subgroups, respectively (*p* < 0.05 for all).

The significant variables from the univariate and multivariate Cox regression model were enrolled to establish the nomogram for predicting the OS of LGG patients (Figure 7b). The concordance index for the model was 0.791 (95%CI = 0.767–0.814). ROC curves and calibration plots were used for evaluating the prediction ability and accuracy of the nomogram. The AUCs for the 1-year, 2-year, and 3-year OS rates with the model were 0.721, 0.693, and 0.666, respectively (Figure 7c). The calibration curves showed a satisfied consistency between the nomogram and the ideal model (Figure 7d).

### 3.8. Validation of Survival Analyses

The prognostic role of ECM2 was further validated using two independent sequencing datasets from the CGGA database. The Cox regression analyses confirmed the independent prognostic role of ECM2 in the CGGA mRNAseq-693 dataset (HR = 2.119, 95%CI = 1.427–3.147, *p* < 0.001, Table 3) and the CGGA mRNAseq-325 dataset (HR = 1.769, 95%CI = 1.049–2.986, *p* = 0.033, Table 4). Kaplan–Meier curves of the CGGA mRNAseq-693 dataset, CGGA mRNAseq-325 dataset, REMBRANDT cohort, and GSE16011 dataset consistently showed a significant correlation between high ECM2 expression and poor prognoses in LGG patients (*p* < 0.05 for all, Figure 8a–d). We confirmed the survival distribution between low and high ECM2 expression in different radiotherapy and chemotherapy subgroups in the CGGA mRNAseq-693 dataset (Figure A2). The results showed that high ECM2 expression was significantly correlated with poor prognosis in the subgroups (*p* < 0.05 for all).

### 3.9. Knockdown of ECM2 Decreased the Proliferation, Migration, and Invasion In Vitro

To further determine the biological function of ECM2, we utilized siRNAs to knockdown ECM2 in U87 and U251 glioma cell lines (Figure 9a–g). After the knockdown of ECM2, we found a significantly decreased ability of wound healing in U87 (#1, *p* = 0.002; #2, *p* = 0.001) and U251 (#1, *p* = 0.001; #2, *p* = 0.011). The proliferation ability was assessed using a CCK-8 assay and a significant difference was found at 72 h (U87 #1, *p* = 0.008; #2, *p* = 0.015) despite no significant difference being found at 24 h or 48 h. Transwell assays showed the migration and invasion ability was consistently decreased in the U87 and U251 cell lines after ECM2 knockdown, indicating the important role of the ECM2 gene in the pathogenesis and development of gliomas.

## 4. Discussion

Gliomas are the most common and deadliest malignant tumors in the central nervous system, with a high recurrence rate and poor prognosis. Although novel treatment strategies, such as immunotherapy, electric field therapy (tumor-treating fields), and oncolytic virus therapy, have garnered increasing attention as promising treatment approaches for gliomas, their current therapeutic effects are limited. Thus, it has been urgent to explore early-stage indicators and prognostic biomarkers to improve the survival of glioma patients. Previous studies have identified the role of ECM2 in cell differentiation and immune regulation [14,25], and recent studies have found its potential role in different types of tumors. However, the role of ECM2 in LGG has hardly been studied. Therefore, we conducted this study to evaluate its expression pattern and prognostic value in gliomas.

In this study, we found that ECM2 expression significantly increased with the grade of glioma, which has been confirmed by IHC results. The results of functional enrichment analyses showed a correlation between ECM2 and immune function. The GSEA results also indicated the potential role of ECM2 in immune regulation. Therefore, we further investigated the association between ECM2 and immune infiltration.

The immune analyses by ssGSEA showed that the infiltration levels were significantly higher in the high ECM2 expression group, and a substantial positive association was observed between ECM2 expression and most immune cells. The TIMER database was used to verify the immune analyses, and significant positive correlations were found between ECM2 expression and immune cells. Based on these results, we consider that the high infiltration levels of macrophages and neutrophils in the high ECM2 expression group may be related to the poor prognoses of patients [26]. The interaction between the glioma and the tumor immune microenvironment (TIM) induces the formation of the immunosuppressive status and promoted tumor progression [27]. Macrophages are abundant in the tumor microenvironment and are associated with chronic inflammation [28]. Tumor cells could promote the transition of macrophages from the M1 to the M2 subtype by secreting cytokines, such as interleukin-4 (IL-4) and transforming growth factor-beta (TGF-β) [29,30]. M2 macrophages could release cytokines and interleukins that facilitate immunological tolerance and tumor proliferation. Interleukin-10 (IL-10) is a potent anti-inflammatory cytokine that suppresses the activity of M1 macrophages and T cells [31]. Furthermore, M2 macrophages secrete vascular endothelial growth factor (VEGF), which promotes angiogenesis and helps to sustain tumor growth [32]. The shift of macrophage phenotype and function within tumors is associated with a poor prognosis in types of tumors [33]. Neutrophils play a double-edged role in the TIM, mediating cytotoxic activities against tumor cells and suppressing metastasis. However, excessive neutrophils can influence the cytolytic activity of NK cells and suppress the proliferation of T cells [34] and contribute to tumor progression by stimulating angiogenesis and inhibiting CD8 T cell activity [26]. We analyzed the survival distribution in different gene expression levels with different immune infiltration by Kaplan–Meier analyses, indicating the combined effect of ECM2 expression and immune cell infiltrating level. Moreover, we found significant positive correlations between most immune checkpoints and ECM2 expression, suggesting the potential role of immunotherapy in patients with high ECM2 expression.

We explored the mutation profiles and found that high ECM2 expression was associated with a lower frequency of IDH1 and IDH2 mutations and a higher frequency of TP53, EGFR, and PTEN mutations. IDH1 and IDH2 mutations are found to be mutually exclusive in gliomas [35], and the presence of mutations in IDH1/IDH2 is associated with a favorable prognosis [36]. In this study, we have found that the LGG patients with low ECM2 expression had a better life expectancy consistent with the higher probability of IDH1/IDH2 mutations. TP53 mutation status has been a critical biomarker for gliomas and about 40% of gliomas exhibit a p53 mutation or deletion [37]. TP53 status plays an important role in chemotherapy resistance to TMZ. The TP53 wild-type glioma cells show a higher sensitivity to TMZ-induced apoptosis than those with mutant TP53 [38,39]. PTEN mutations have been studied to be correlated with poor prognoses in glioma [40]. PTEN mutations are shown to be a late event in the progression of glioma [41]. Previous studies have identified the critical role of EGFR mutation and amplification in tumorigenesis [42]. EGFRvIII is the most common EGFR mutation in glioma [43]. Although current studies have shown that lung cancer patients could benefit from EGFR tyrosine kinase inhibitors (TKIs), the therapeutic effect of EGFR TKIs in glioma remains unclear.

We found that the expression of ECM2 was significantly higher in WHO grade 3, IDH wild-type, and 1p/19q non-codeletion subtypes, which suggested that high ECM2 expression was a potential biomarker for a poor prognosis in LGGs. Then, we performed the Cox regression analyses and identified ECM2 expression as an independent prognostic indicator for LGG patients, and Kaplan–Meier analyses showed a significant difference in survival distribution between low and high ECM2 expression groups. The nomogram model was established and its reliable predictive ability has been assessed. To verify the prognostic role of ECM2, a total of 894 LGG samples from four external datasets have been included. The CGGA mRNAseq-693 dataset and CGGA mRNAseq-325 dataset have validated the prognostic role of ECM2, respectively. The survival distribution between low and high ECM2 expression groups has been validated by the CGGA mRNAseq-693 dataset, CGGA mRNAseq-325 dataset, REMBRANDT cohort, and GSE16011 dataset. These results suggested that ECM2 expression is an independent prognostic factor for LGG patients. Specifically, patients with high ECM2 expression show significantly shorter OS than those with low ECM2 expression. This is consistent with the results of in vitro functional experiments that high ECM2 expression promotes the proliferation, migration, and invasiveness of glioma cells.

The ECM is an essential component of the tumor microenvironment. One of the critical functions of the ECM is to provide a physical scaffold that supports tumor growth and angiogenesis [44]. ECM can also regulate tumor cell behavior through its interactions with cell-surface receptors, which promote the survival, proliferation, migration, and invasiveness of tumor cells [45]. Moreover, the tumor-associated ECM can induce immunosuppressive effects by promoting the recruitment of immune-suppressive cells and inhibiting the activation of cytotoxic T cells [46]. As a component of ECM, the research on ECM2 is relatively limited. In this study, we found that ECM2 is significantly increased in gliomas and it is associated with the immune regulation in TIM. The survival analyses identified its prognostic value in LGG patients and the results of in vitro experiments indicated its role in the proliferation, migration, and invasiveness of glioma cells. Our study provides a novel sight into understanding the role of ECM2 in glioma, which contributes to the development of new therapeutic strategies for glioma patients.

However, our study still had some limitations. First, this is a retrospective study, and the research on the prognostic role of ECM2 is based on the open data of public datasets. Currently, these datasets lack clinically relevant information on the scope of resection. We will conduct prospective cohort studies in the following studies to verify the prognostic effect of ECM2 and include information on the scope of resection. Second, further studies are needed to understand the downstream signaling pathways in glioma and the regulatory mechanisms in immune infiltration. Although the in vitro experiments could verify the role of ECM2 in glioma cell lines, further in vivo experiments are necessary to reveal the effect of ECM2 on the pathogenesis and progression of glioma.

## 5. Conclusions

In summary, our study found that high ECM2 expression was associated with a poor prognosis in LGG patients and suggested that ECM2 could serve as a novel immune-associated prognostic indicator for LGG patients and a potential therapeutic target for immunotherapy.

## Figures and Tables

**Figure 1 brainsci-13-00851-f001:**
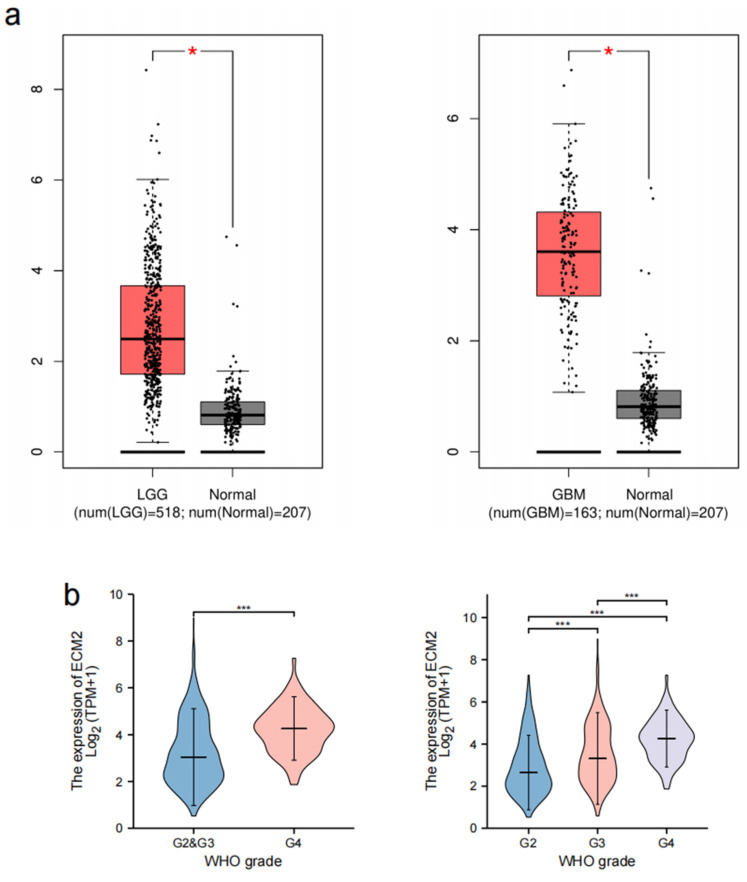
Expression patterns of ECM2 in gliomas. (**a**) Expression of ECM2 between LGG, GBM samples, and normal tissues. (**b**) Expression of ECM2 in LGG and GBM, and WHO grade 2, 3, and 4 gliomas. (**c**) Immunohistochemical analyses of ECM2 expression in low-grade and high-grade gliomas. ns, not significant; *, *p* < 0.05; ***, *p* < 0.001.

**Figure 2 brainsci-13-00851-f002:**
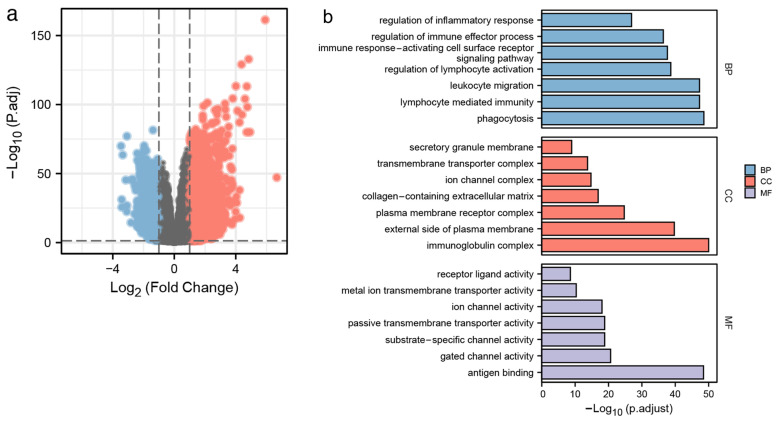
Identification of DEGs, *ECM2*-associated gene network, and functional enrichment analyses. (**a**) Volcano plot of DEGs between low and high *ECM2* expression groups. (**b**) GO enrichment analyses. (**c**) KEGG pathway annotation. (**d**) Genetic interaction network of *ECM2*. (**e**) GSEA analyses.

**Figure 3 brainsci-13-00851-f003:**
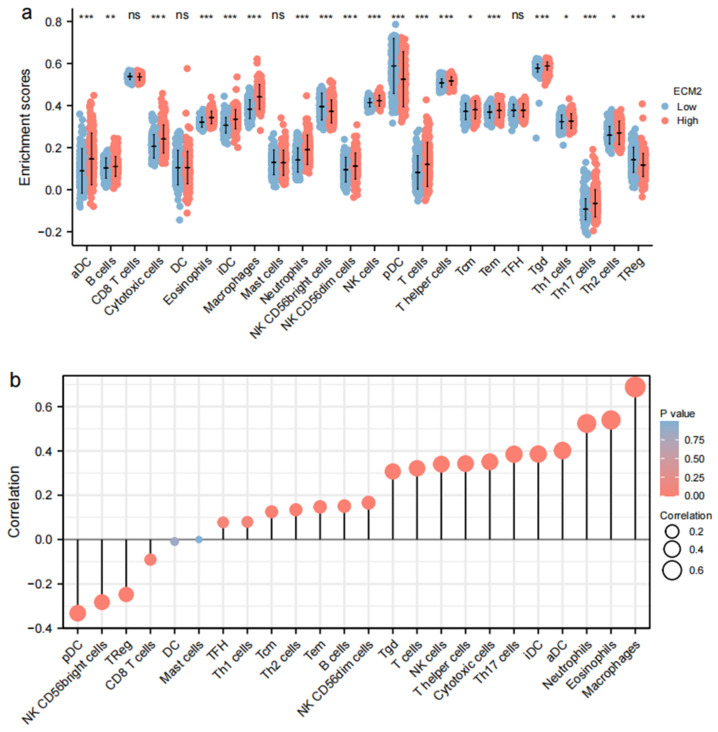
Correlation of *ECM2* expression with immune infiltration and immune checkpoints. (**a**) Infiltration of immune cells between low and high *ECM2* expression groups. (**b**) Correlation between *ECM2* expression and immune cell infiltration. (**c**) Correlation of *ECM2* expression with immune cell infiltration by TIMER. (**d**) Cumulative survival analyses of *ECM2* expression and immune cell infiltration. (**e**) Heat map of immune checkpoint expression. ns, not significant; *, *p* < 0.05; **, *p* < 0.01; ***, *p* < 0.001.

**Figure 4 brainsci-13-00851-f004:**
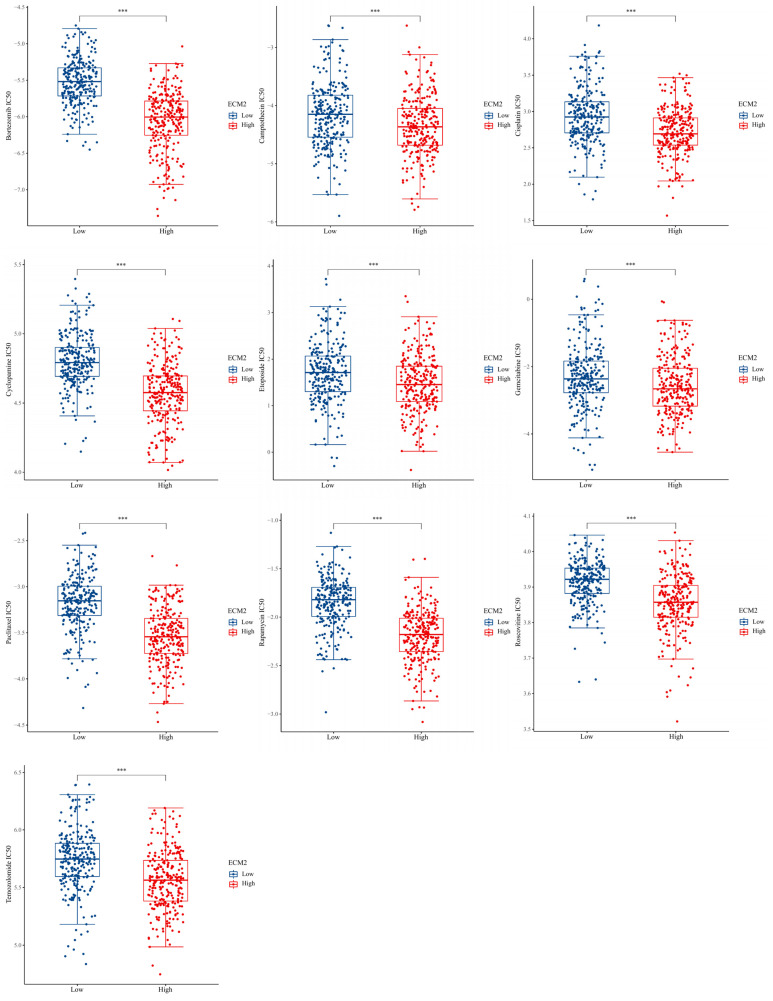
Drug sensitivity analysis between low and high *ECM2* expression groups, *** *p* < 0.001.

**Figure 5 brainsci-13-00851-f005:**
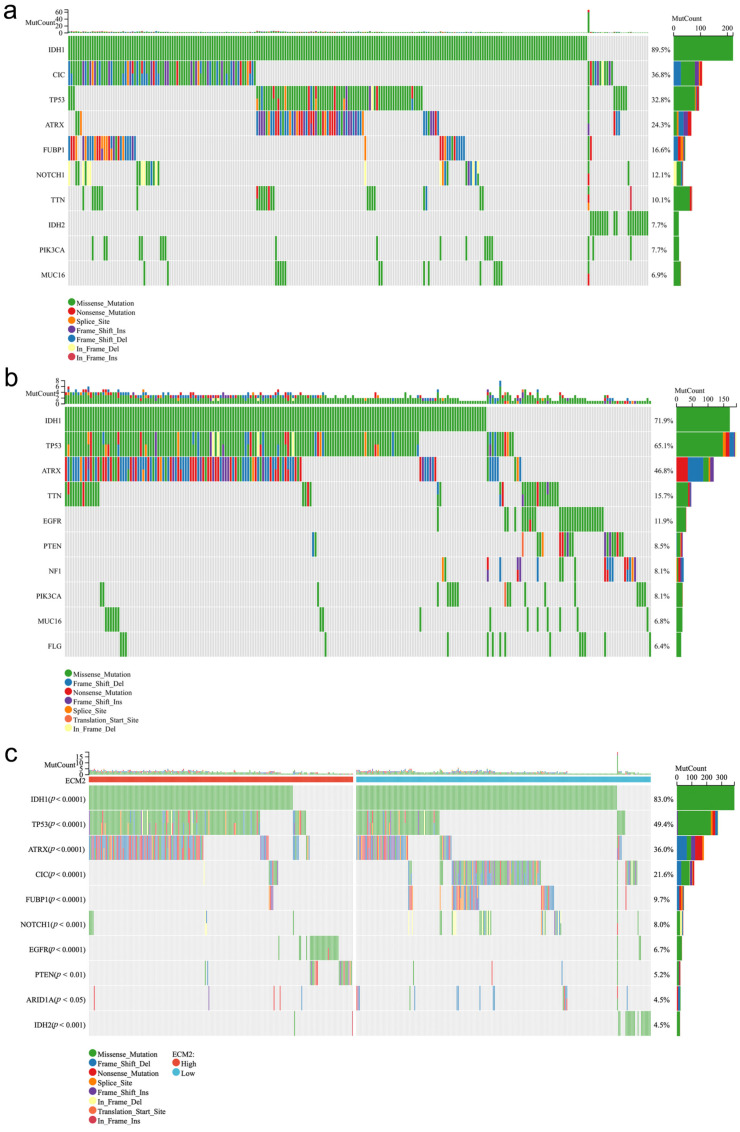
Mutation profiles of low and high *ECM2* expression groups. (**a**) The frequencies of the top-10 mutated genes in low the *ECM2* expression group. (**b**) The frequencies of the top-10 mutated genes in the high *ECM2* expression group. (**c**) Comparison of the frequencies of the top-10 mutant genes in LGGs between low and high *ECM2* expression groups.

**Figure 6 brainsci-13-00851-f006:**
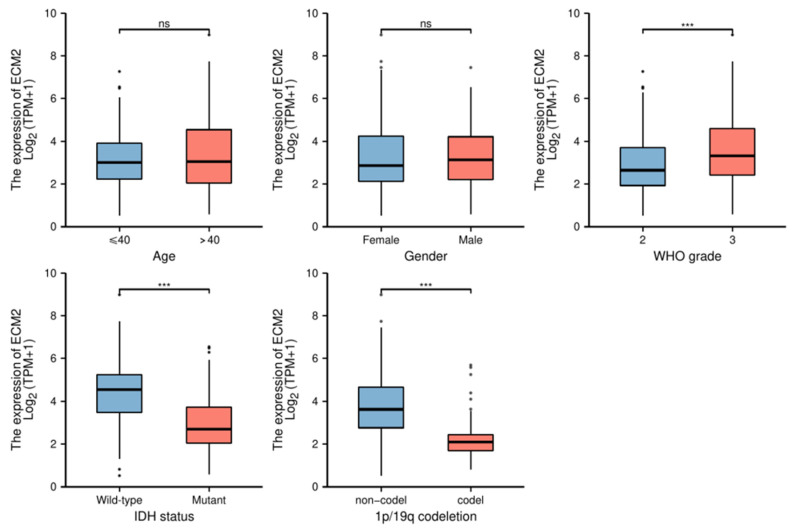
Expression of *ECM2* in different clinical subgroups, including age, gender, WHO grade, IDH status, and 1p/19q codeletion status. Dots, outliers; ns, not significant; ***, *p* < 0.001.

**Figure 7 brainsci-13-00851-f007:**
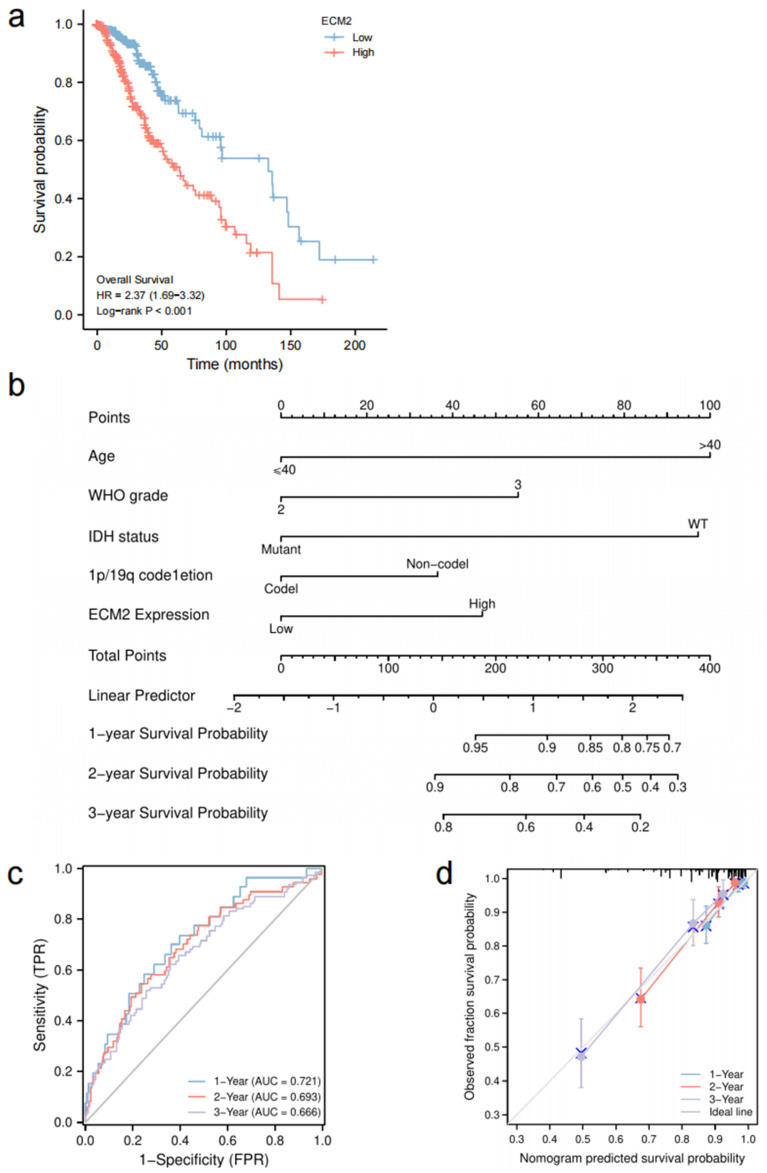
Survival analysis and prognosis prediction model for LGG patients. (**a**) Kaplan–Meier survival analysis between low and high *ECM2* expression groups. (**b**) Nomogram for 1-, 2-, and 3-year OS rates. (**c**) Time-dependent ROC curves. (**d**) Calibration plots of the nomogram.

**Figure 8 brainsci-13-00851-f008:**
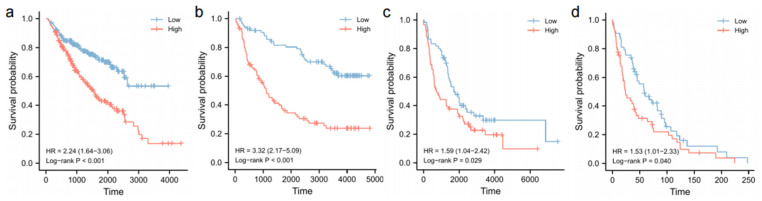
Kaplan–Meier survival analyses and log-rank tests in different validation datasets. (**a**) CGGA mRNAseq-693 dataset; (**b**) CGGA mRNAseq-325 dataset; (**c**) REMBRANDT dataset; (**d**) GSE16011 dataset.

**Figure 9 brainsci-13-00851-f009:**
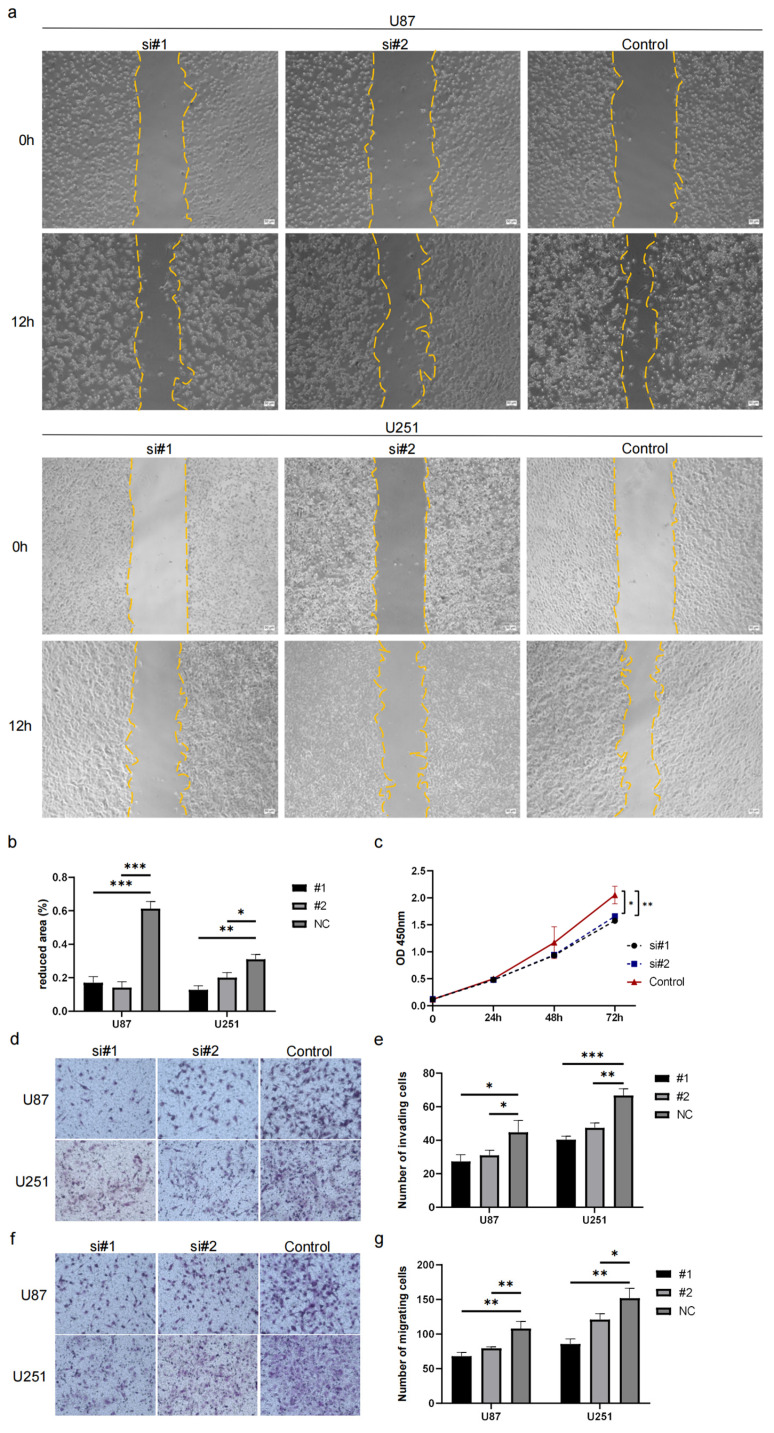
Biological function of *ECM2* in glioma U87 and U251 cell lines. (**a**,**b**) Wound healing assays; (**c**) CCK-8 assays; (**d**,**e**) Transwell assays, invasion assays; (**f**,**g**) Transwell assays, migration assays. *, *p* < 0.05; **, *p* < 0.01; ***, *p* < 0.001.

**Table 1 brainsci-13-00851-t001:** Clinicopathologic features of low and high *ECM2* expression groups in LGG patients from TCGA database.

Characteristic	Low Expression of *ECM2*	High Expression of *ECM2*	*p* Value
Age, *n* (%)			0.931
≤40	133 (50.4%)	131 (49.6%)	
>40	131 (49.6%)	133 (50.4%)	
Gender, *n* (%)			0.162
Female	128 (48.5%)	111 (42%)	
Male	136 (51.5%)	153 (58%)	
WHO grade, *n* (%)			<0.001 *
2	165 (31.2%)	120 (22.7%)	
3	99 (18.8%)	144 (61.3%)	
IDH status, *n* (%)			<0.001 *
Wild type	14 (5.3%)	86 (16.3%)	
Mutant	250 (94.7%)	178 (68.2%)	
1p/19q codeletion, *n* (%)			<0.001 *
Noncodel	110 (41.7%)	247 (93.6%)	
Codel	154 (58.3%)	17 (6.4%)	

** p* < 0.05, significant difference.

**Table 2 brainsci-13-00851-t002:** Univariate and multivariate Cox analyses in TCGA dataset.

Characteristic	Univariate Analysis		Multivariate Analysis	
	Hazard Ratio (95%CI)	*p* Value	Hazard Ratio (95%CI)	*p* Value
Age				
≤40	Reference		Reference	
>40	2.889 (2.009–4.155)	<0.001 *	2.931 (1.989–4.320)	<0.001 *
Gender				
Female	Reference			
Male	1.124 (0.800–1.580)	0.499		
WHO grade				
2	Reference		Reference	
3	2.354 (1.664–3.332)	<0.001 *	1.812 (1.266–2.593)	0.001 *
IDH status				
Wild type	Reference		Reference	
Mutant	0.206 (0.145–0.292)	<0.001 *	0.351 (0.236–0.523)	<0.001 *
1p/19q codeletion				
Noncodel	Reference		Reference	
Codel	0.401 (0.256–0.629)	<0.001 *	0.676 (0.380–1.199)	0.181
Radiotherapy				
No	Reference			
Yes	0.950 (0.610–1.480)	0.820		
Chemotherapy				
No	Reference			
Yes	1.415 (0.877–2.284)	0.155		
*ECM2* expression				
Low	Reference		Reference	
High	2.449 (1.698–3.534)	<0.001 *	1.656 (1.055–2.600)	0.028 *

* *p* < 0.05, significant difference.

**Table 3 brainsci-13-00851-t003:** Univariate and multivariate Cox analyses in CGGA mRNAseq-693 dataset.

Characteristic	Univariate Analysis		Multivariate Analysis	
	Hazard Ratio (95%CI)	*p* Value	Hazard Ratio (95%CI)	*p* Value
Age				
≤40	Reference			
>40	1.208 (0.882–1.655)	0.239		
Gender				
Female	Reference			
Male	1.007 (0.734–1.381)	0.965		
WHO grade				
2	Reference		Reference	
3	2.544 (1.780–3.635)	<0.001 *	2.673 (1.787–4.000)	<0.001 *
IDH status				
Wild type	Reference		Reference	
Mutant	0.459 (0.325–0.647)	<0.001 *	0.581 (0.390–0.867)	0.008 *
1p/19q codeletion				
Noncodel	Reference		Reference	
Codel	0.353 (0.231–0.541)	<0.001 *	0.562 (0.347–0.911)	0.019 *
Radiotherapy				
No	Reference		Reference	
Yes	1.592 (1.012–2.505)	0.044 *	1.352 (0.818–2.233)	0.239
Chemotherapy				
No	Reference			
Yes	1.290 (0.889–1.873)	0.180		
*ECM2* expression				
Low	Reference		Reference	
High	2.408 (1.734–3.343)	<0.001 *	2.119 (1.427–3.147)	<0.001 *

** p* < 0.05, significant difference.

**Table 4 brainsci-13-00851-t004:** Univariate and multivariate Cox analyses in CGGA mRNAseq-325 dataset.

Characteristic	Univariate Analysis		Multivariate Analysis	
	Hazard Ratio (95%CI)	*p* Value	Hazard Ratio (95%CI)	*p* Value
Age				
≤40	Reference			
>40	1.348 (0.891–2.039)	0.157		
Gender				
Female	Reference		Reference	
Male	0.635 (0.420–0.960)	0.031 *	0.624 (0.401–0.968)	0.036 *
WHO grade				
2	Reference		Reference	
3	3.579 (2.335–5.485)	<0.001 *	3.265 (1.975–5.397)	<0.001 *
IDH status				
Wild type	Reference		Reference	
Mutant	0.379 (0.244–0.587)	<0.001 *	0.869 (0.532–1.420)	0.576
1p/19q codeletion				
Noncodel	Reference		Reference	
Codel	0.156 (0.082–0.294)	<0.001 *	0.233 (0.110–0.492)	<0.001 *
Radiotherapy				
No	Reference		Reference	
Yes	0.564 (0.339–0.938)	0.027 *	0.773 (0.445–1.340)	0.359
Chemotherapy				
No	Reference		Reference	
Yes	1.610 (1.039–2.495)	0.033 *	1.101 (0.673–1.803)	0.702
*ECM2* expression				
Low	Reference		Reference	
High	3.451 (2.211–5.385)	<0.001 *	1.769 (1.049–2.986)	0.033 *

** p* < 0.05, significant difference.

## Data Availability

We have presented all the public databases used for analyses in the manuscript. The expression between LGG, GBM samples, and normal tissues was compared by the GEPIA platform (http://gepia.cancer-pku.cn/index.html). The RNA-seq and the matching clinical data for analyses were obtained from the TCGA database (https://portal.gdc.cancer.gov). The immunohistochemical analyses were obtained from Human Protein Atlas (http://www.proteinatlas.org/). The gene network analyses were obtained by the GeneMANIA database (http://www.genemania.org). The immune infiltration analyses were validated by the TIMER database (https://cistrome.shinyapps.io/timer/). The gene expression data and clinical features for validation were obtained from the CGGA database (http://www.cgga.org.cn/), REMBRANDT cohort (http://www.betastasis.com/glioma/rembrandt/), and the GSE16011 dataset (https://www.ncbi.nlm.nih.gov/gds/) all accessed on 20 November 2022.

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
