# Peer review of "High *ECM2* Expression Predicts Poor Clinical Outcome and Promotes the Proliferation, Migration, and Invasiveness of Glioma"

_brainsci, 2023, doi:10.3390/brainsci13060851_

Round 1
Reviewer 1 Report (Previous Reviewer 1)
The authors addressed all issues. In my opinion, the manuscript can be considered for publication.
Author Response
Thanks for your careful review. We really appreciate your efforts on our paper.
Reviewer 2 Report (New Reviewer)
High ECM2 Expression Predicts Poor Clinical Outcome and Promotes Proliferation, Migration and Invasiveness of Glioma, the manuscript is interesting and has a big potential however, some sections of the MS are misleading. Please see my comments
Introduction:
"Glioma was a highly aggressive neoplastic disease" I think you should change the past tense to present tense in this sentence and the following.
I would suggest to develop the introduction and provide more background, the hypothesis and the aims of the study please.
Materials and methods:
Please develop the 2.1 section
2.3 please develop the technique
for analyses on human do you have an ethic protocol and consent or it's from a biobank ?
Results
Fig 1a, please add the corresponding legend on the graphs. What is the orange, blue. It would help to clarify
Fig 1b on the right, on the x axis you write grade 4, but on this other graph you call it GBM if I well understand. To keep the writing homogeneous I would suggest to change it for GBM or GBM for 4
Result 3.3, from the material and method to the 3.2 section, it's not clear how you generated these results. What is the cohort, how did you choose it, describe the demographic characteristic of the cohort. It is misleading without it. You provide such data later in the MS, so please clarify that.
same comment for 3.4 and 3.5.
3.5 figures are too small and are hard to read
"We further analyzed the survival distribution between low and high ECM2 expres- 285 sion in different radiotherapy and chemotherapy subgroups (Figure A1). The results 286 showed that a significant correlation between high ECM2 expression and poor prognosis 287 in the subgroups, respectively (P<0.05 for all). 288 The significant variables from the univariate and multivariate Cox regression model 289 were enrolled to establish the nomogram for predicting the OS of LGG patients (Figure 290 7b)." this is highlighted in red .... same for line 310 to 315
Figure 9, images and graphs are really small it's hard to read, please adjust the size.
Discussion:
"Glioma cells could stimulate the transition of macrophages from M1 to M2 subtype 364 [29], and macrophages M2 could release cytokines and interleukins, which facilitated 365 immunological tolerance and tumor proliferation [30]." Please demonstrate this point. It would greatly help to understand the mechanism.
"These results supported the use of ECM2 as a reliable prognostic bi- 412 omarker for LGG patients" Please be more specific.
In the discussion, I would suggest you to develop it in light of the literature, also please add the perspective, why this work would help the science and add the limitations.
Some typos here and there, and the use of past should be modified in the introduction and discussion
Author Response
Responds to the reviewer’s comments:
Reviewer #2:
General comment:
High ECM2 Expression Predicts Poor Clinical Outcome and Promotes Proliferation, Migration and Invasiveness of Glioma, the manuscript is interesting and has a big potential however, some sections of the MS are misleading.
Reply:
Thank you for your insightful comment. We appreciate your professional opinion and expertise.
1. Comment:
"Glioma was a highly aggressive neoplastic disease" I think you should change the past tense to present tense in this sentence and the following.
1. Reply:
Thank you very much for your valuable comment. We have changed the past tense to present tense. The revised content has been shown.
“Glioma is a highly aggressive neoplastic disease originating from glial cells and is considered as the most common type of brain tumor [1]. According to the World Health Organization (WHO) classification, grade 2 and 3 gliomas are referred to lower-grade glioma (LGG), while grade 4 is known as glioblastoma (GBM) [2]. Although LGG shows a more indolent behavior and slower growth than GBM, the local recurrence and malignant transformation seem to be inevitable due to the infiltrating growth pattern [3]. As a result of glioma heterogeneity, patients have different sensitivity to chemotherapy and radiotherapy. Despite the emergence of new treatment options, such as targeted therapy and immunotherapy, the therapeutic efficacy remains limited and the overall survival (OS) of patients varies widely [4-6]. Thus, there has been an urgent need to identify novel biomarkers for prognostic prediction and individualized treatment in glioma patients.”
2. Comment:
I would suggest to develop the introduction and provide more background, the hypothesis and the aims of the study please.
2. Reply:
Thank you so much for your nice suggestion. We have developed the section of Introduction and added the hypothesis and the aims of the study.
“Increasing evidence has shown that extracellular matrix (ECM) plays an important role in pathogenesis, development, and therapy of tumors. Extracellular Matrix Protein 2 (ECM2) is located on chromosomal 9q22.3 [7]. As an extracellular matrix component, ECM2 is involved in the regulation of cell proliferation and differentiation [8]. Recent studies have found the association between ECM2 and colon adenocarcinoma, hepatocellular carcinoma, and cervical cancer [9-11]. Additionally, ECM2 has been considered to be associated with idiopathic pulmonary fibrosis and pulmonary artery hypertension [12,13]. A previous study reveals the association between ECM2 and lymphopoiesis that ECM2 enhances the proliferation of mature B cells [14]. Therefore, we consider that ECM2 plays a potential role in tumor immune microenvironment and affects the prognoses in LGG patients. However, there are few studies focused on the role of ECM2 in gliomas and its prognostic value for LGG patients is still unclear.
In this study, we investigate the role of ECM2 in gliomas and its prognostic value for LGG patients. We perform the functional enrichment analyses, immune analyses, and drug sensitivity of ECM2 to identify its potential function. We further identify and verify its prognostic value using four external datasets. The function of ECM2 in glioma is confirmed with in vitro experiments. Overall, our study suggests that ECM2 is a promising prognostic indicator and potential therapeutic target for glioma patients.”
3. Comment:
Please develop the 2.1 section, 2.3 please develop the technique.
3. Reply:
Thanks for your nice suggestion. We have developed the section of 2.1 and 2.3. The revised content has been shown.
“2.1. Gene expression analysis in gliomas
The GEPIA platform (http://gepia.cancer-pku.cn/index.html), an interactive platform for analyzing RNA sequencing expression data of tumors and normal samples from the TCGA and GTEx projects, was used to compare the ECM2 expression levels of LGG and GBM samples with that of normal tissues. The level 3 HTSeq-count data and corresponding clinical features of 529 LGG samples were obtained from TCGA database. Further analyses were conducted to compare the expression levels between different grades of glioma. The protein expression levels of ECM2 between different grades of glioma were confirmed by immunohistochemical (IHC) staining from Human Protein Atlas (HPA) database (http://www.proteinatlas.org/). We performed the functional enrichment analyses, immune-related analyses, mutant profiles, and drug sensitivity analyses based on TCGA database.
2.3. Immune-related analyses
The immune infiltration analyses were performed by single-sample Gene Set Enrichment Analysis (ssGSEA) with GSVA package [19]. The levels of 24 types of infiltrating immune cells were compared between low and high ECM2 expression groups [20]. The correlation between ECM2 expression and immune cells was also analyzed using Spearman correlation tests. The immune infiltration analyses were validated by TIMER database, a comprehensive resource for systematic immune infiltration estimation with 10,897 samples across 32 cancer types from TCGA [21]. Furthermore, Kaplan-Meier plots were performed to evaluate the combined effect of ECM2 expression level and immune infiltration on survival probabilities. The correlations between the expression of ECM2 and documented immune checkpoints were also analyzed using Spearman correlation tests [22], and the expression of immune checkpoints was displayed using a heat map by ggplot2 package.”
4. Comment:
for analyses on human do you have an ethic protocol and consent or it's from a biobank ?
4. Reply:
Thanks for your rigorous consideration. We have added the Ethics series number and revised the manuscript as following.
“This study has been reviewed and approved by the Ethics Committee in Beijing Tiantan Hospital, Capital Medical University (KY2022-178-02).”
5. Comment:
Fig 1a, please add the corresponding legend on the graphs. What is the orange, blue. It would help to clarify
5. Reply:
Thanks for your nice suggestion. We have revised the Figure 1a and the revised figure has been shown below.
6. Comment:
Fig 1b on the right, on the x axis you write grade 4, but on this other graph you call it GBM if I well understand. To keep the writing homogeneous I would suggest to change for GBM or GBM for 4
6. Reply:
Thanks for your nice suggestion. We have revised the Figure 1b and the revised figure has been shown below.
7. Comment:
Result 3.3, from the material and method to the 3.2 section, it's not clear how you generated these results. What is the cohort, how did you choose it, describe the demographic characteristic of the cohort. It is misleading without it. You provide such data later in the MS, so please clarify that. same comment for 3.4 and 3.5.
7. Reply:
Thanks for your rigorous consideration. We generated the results with the RNA-seq expression profile of the LGG samples from TCGA. In addition to the validation of the prognostic role of ECM2, all analyses were based on the TCGA database. The methods were consistent with our previous studies [1-2]. We have added this in the section of Method 2.1. “We performed the functional enrichment analyses, immune-related analyses, mutant profiles, and drug sensitivity analyses based on TCGA database.” The demographic characteristics between low and high ECM2-expression groups were shown in the section of Results (3.6. Association between clinical features and ECM2 expression). The table of demographic and clinical characteristics has been shown below.
- Li J, Wang J, Liu D, Tao C, Zhao J, Wang W. Establishment and validation of a novel prognostic model for lower-grade glioma based on senescence-related genes. Front Immunol. 2022 Oct 21;13:1018942. doi: 10.3389/fimmu.2022.1018942.
- Li J, Wang J, Ding Y, Zhao J, Wang W. Prognostic biomarker SGSM1 and its correlation with immune infiltration in gliomas. BMC Cancer. 2022 Apr 28;22(1):466. doi: 10.1186/s12885-022-09548-7.
8. Comment:
3.5 figures are too small and are hard to read
8. Reply:
Thanks for your rigorous consideration. We have revised the font size and dpi of Figure 5, and the revised figures have been shown below.
9. Comment:
"We further analyzed the survival distribution between low and high ECM2 expres- 285 sion in different radiotherapy and chemotherapy subgroups (Figure A1). The results 286 showed that a significant correlation between high ECM2 expression and poor prognosis 287 in the subgroups, respectively (P<0.05 for all). 288 The significant variables from the univariate and multivariate Cox regression model 289 were enrolled to establish the nomogram for predicting the OS of LGG patients (Figure 290 7b)." this is highlighted in red .... same for line 310 to 315
9. Reply:
Thanks for your rigorous consideration. Sorry for this mistake in edition and the inconvenience it caused in your reading. We have removed the highlight of the relevant content.
10. Comment:
Figure 9, images and graphs are really small it's hard to read, please adjust the size.
10. Reply:
Thanks for your nice suggestion. We have revised the font size and dpi of Figure 9, and the revised figure has been shown below.
11. Comment:
"Glioma cells could stimulate the transition of macrophages from M1 to M2 subtype 364 [29], and macrophages M2 could release cytokines and interleukins, which facilitated 365 immunological tolerance and tumor proliferation [30]." Please demonstrate this point. It would greatly help to understand the mechanism.
11. Reply:
Thanks for your nice suggestion. We have revised the relevant content in the manuscript.
“Tumor cells could promote the transition of macrophages from M1 to M2 subtype by secreting cytokines, such as interleukin-4 (IL-4) and transforming growth factor-beta (TGF-β) [29,30]. M2 macrophages could release cytokines and interleukins that facilitate immunological tolerance and tumor proliferation. Interleukin-10 (IL-10) is a potent anti-inflammatory cytokine that suppresses the activity of M1 macrophages and T cells [31]. Furthermore, M2 macrophages secrete vascular endothelial growth factor (VEGF), which promotes angiogenesis and helps to sustain tumor growth [32]. The shift of macrophage phenotype and function within tumors are associated with a poor prognosis in types of tumors [33].”
12. Comment:
"These results supported the use of ECM2 as a reliable prognostic bi- 412 omarker for LGG patients" Please be more specific.
12. Reply:
Thanks for your nice suggestion. We have revised the relevant content in the manuscript.
“These results suggested that ECM2 expression is an independent prognostic factor for LGG patients. Specifically, patients with high ECM2 expression show significantly shorter OS than those with low ECM2 expression. This is consistent with the results of in vitro functional experiments that high ECM2 expression promotes proliferation, migration and invasiveness of glioma cells.”
13. Comment:
In the discussion, I would suggest you to develop it in light of the literature, also please add the perspective, why this work would help the science and add the limitations.
13. Reply:
Thanks for your nice suggestion. We have revised the relevant content in the manuscript.
“The ECM is an essential component of the tumor microenvironment. One of the critical functions of the ECM is to provide a physical scaffold that supports tumor growth and angiogenesis [44]. ECM can also regulate tumor cell behavior through its interactions with cell surface receptors, which promote the survival, proliferation, migration, and invasiveness of tumor cells [45]. Moreover, the tumor-associated ECM can induce immunosuppressive effects by promoting the recruitment of immune suppressive cells and inhibiting the activation of cytotoxic T cells [46]. As a component of ECM, the research on ECM2 is relatively limited. In this study, we found that ECM2 is significantly increased in gliomas and it is associated with the immune regulation in TIM. The survival analyses identified its prognostic value in LGG patients and the results of in vitro experiments indicated its role in proliferation, migration and invasiveness of glioma cells. Our study provides a novel sight in understanding the role of the ECM2 in glioma, which contributes to the development of new therapeutic strategies for glioma patients.
However, our study still had some limitations. First, this is a retrospective study, and the research on the prognostic role of ECM2 is based on the open data of the public datasets. Currently, these datasets lack clinically relevant information on the scope of resection. We will conduct prospective cohort studies in following studies to verify the prognostic effect of ECM2 and include information on the scope of resection. Second, further studies are needed to understand the downstream signaling pathways in glioma and the regulatory mechanisms in immune infiltration. Although the in vitro experiments could verify the role of ECM2 in glioma cell lines, further in vivo experiments are necessary to reveal the effect of ECM2 on the pathogenesis and progression of glioma.”
14. Comment:
Some typos here and there, and the use of past should be modified in the introduction and discussion.
14. Reply:
Thanks for your valuable comment. We seriously considered your nice suggestion and revised our manuscript thoroughly. The typos and the past tense have been revised.
Thanks for your careful review. We really appreciate your efforts on our paper and hope that the response will meet with approval. Your careful review has helped to make our study clearer and more comprehensive.

Round 2
Reviewer 2 Report (New Reviewer)
thanks for the modifications
This manuscript is a resubmission of an earlier submission. The following is a list of the peer review reports and author responses from that submission.
Round 1
Reviewer 1 Report
The authors present an interesting study regarding the prognostic role of ECM2 expression in glioma
- Introduction: The authors should adhere to the present nomenclature of the WHO classification system. Arabic numerals should be used and the term GBM does no more exist. In lines 63-64 the authors describe “few studies focused on the role of ECM2 in gliomas”. Please add the references. I do not understand the lines 65-70 in the section discussion. The authors describe results and conclusions of the study in this paragraph.
- Methods & Results: I cannot agree with the definition of a low-grade glioma. The current WHO classification grades IDH-1 wild-type gliomas as WHO grade 4 tumors. Furthermore, table 1 shows a significant increased proportion of IDH-1 wild type “low grade gliomas” among those with an increased expression of ECM2. Hence, I think the identified prognostic role of increased expression of ECM2 in overall survival is predominantly based on the wrong definition of the term low-grade glioma in this study. Furthermore, the multivariable cox regression analysis does not consider relevant clinical predictors of overall survival (e.g. extent of resection, adjuvant radiochemotherapy completion/interruption).
All in all, the present study addresses a very important objective and provides a lot of data. However, I think there are some major methodological errors.
Author Response
Responds to the reviewer’s comments:
Reviewer #1:
General Comments: The authors present an interesting study regarding the prognostic role of ECM2 expression in glioma
Reply: Thank you very much for your valuable comment. We really appreciate your efforts on our manuscript.
1. Comment: Introduction: The authors should adhere to the present nomenclature of the WHO classification system. Arabic numerals should be used and the term GBM does no more exist. In lines 63-64 the authors describe“few studies focused on the role of ECM2 in gliomas”. Please add the references. I do not understand the lines 65-70 in the section discussion. The authors describe results and conclusions of the study in this paragraph.
1. Reply: Thank you very much for your valuable comment. We really appreciate your efforts on our manuscript. This was a retrospective study.The classification of all samples in TCGA and CGGA databases was based on the WHO classification at the corresponding time, and such expression is widely used in recent articles [1-5]. Our study is the first one focusing on ECM2 and its function in gliomas, which has been verified by cell experiments. According to your comments, we have removed the expression of “few studies focused on the role of ECM2 in gliomas”. We aimed to highlight the importance of our findings in predicting the prognosis of glioma patients in the section of Discussion, and we have now revised the relevant content in the manuscript. Thanks again for your professional opinion.
- Lu J, Peng Y, Huang R, Feng Z, Fan Y, Wang H, Zeng Z, Ji Y, Wang Y, Wang Z. Elevated TYROBP expression predicts poor prognosis and high tumor immune infiltration in patients with low-grade glioma. BMC Cancer. 2021 Jun 23;21(1):723. doi: 10.1186/s12885-021-08456-6.
- Hu M, Li Z, Qiu J, Zhang R, Feng J, Hu G, Ren J. CKS2 (CDC28 protein kinase regulatory subunit 2) is a prognostic biomarker in lower grade glioma: a study based on bioinformatic analysis and immunohistochemistry. Bioengineered. 2021 Dec;12(1):5996-6009. doi: 10.1080/21655979.2021.1972197.
- Yu K, Ji Y, Liu M, Shen F, Xiong X, Gu L, Lu T, Ye Y, Feng S, He J. High Expression of CKS2 Predicts Adverse Outcomes: A Potential Therapeutic Target for Glioma. Front Immunol. 2022 May 19;13:881453. doi: 10.3389/fimmu.2022.881453.
- Zhang Z, Liu S. The interaction between ASF1B and TLK1 promotes the malignant progression of low-grade glioma. Ann Med. 2023 Dec;55(1):1111-1122. doi: 10.1080/07853890.2023.2169751.
- Li T, Yang Z, Li H, Zhu J, Wang Y, Tang Q, Shi Z. Phospholipase Cγ1 (PLCG1) overexpression is associated with tumor growth and poor survival in IDH wild-type lower-grade gliomas in adult patients. Lab Invest. 2022 Feb;102(2):143-153. doi: 10.1038/s41374-021-00682-7.
2. Comment: Methods & Results: I cannot agree with the definition of a low-grade glioma. The current WHO classification grades IDH-1 wild-type gliomas as WHO grade 4 tumors. Furthermore, table 1 shows a significant increased proportion of IDH-1 wild type “low grade gliomas”among those with an increased expression of ECM2. Hence, I think the identified prognostic role of increased expression of ECM2 in overall survival is predominantly based on the wrong definition of the term low-grade glioma in this study. Furthermore, the multivariable cox regression analysis does not consider relevant clinical predictors of overall survival (e.g. extent of resection, adjuvant radiochemotherapy completion/interruption).
2. Reply: Thank you very much for your valuable comment. Our definition was lower-grade glioma rather than low-grade glioma which you mentioned. Thisdefinitionwas consistent with the expression in recent studies [6-10]. The clinical variables included for Cox regression were the same as previous studies [11-13]. We carefully searched the TCGA and CGGA databases, but did not find any information about the extent of resection. The information of radiotherapy and chemotherapy in TCGA database was severely incomplete. Therefore, we further included the information of radiotherapy and chemotherapy data in CGGA mRNAseq-693 dataset and CGGA mRNAseq-325 dataset to explore the prognostic role of ECM2. The results showed that ECM2 was still independently associated with the prognosis of LGG patients, even after the inclusion of radiotherapy and chemotherapy. The results have been shown below. Thanks again for your careful review. We really appreciate your efforts on our paper and hope that the response will meet with approval. Your careful review has helped to make our study clearer and more comprehensive.
- Maimaiti A, Feng Z, Liu Y, Turhon M, Xie Z, Baihetiyaer Y, Wang X, Kasimu M, Jiang L, Wang Y, Wang Z, Pei Y. N7-methylguanosin regulators-mediated methylation modification patterns and characterization of the immune microenvironment in lower-grade glioma. Eur J Med Res. 2023 Mar 30;28(1):144. doi: 10.1186/s40001-023-01108-4.
- Huang X, Vafaei S, Li L, Wang Y, Sun J, Gao Y, Zhang J. Identification of necroptosis-related genes as prognostic indicators for lower-grade glioma. Am J Cancer Res. 2023 Feb 15;13(2):692-708.
- Aili Y, Maimaitiming N, Maimaiti A, Liu W, Qin H, Ji W, Mahemuti Y, Wang Y, Wang Z. Identification of VASH1 as a Potential Prognostic Biomarker of Lower-Grade Glioma by Quantitative Proteomics and Experimental Verification. J Oncol. 2022 Nov 30;2022:2621969. doi: 10.1155/2022/2621969.
- Wu H, He H, Huang J, Wang C, Dong Y, Lin R, Cheng Z, Qiu Q, Hong L. Identification and validation of transferrin receptor protein 1 for predicting prognosis and immune infiltration in lower grade glioma. Front Mol Neurosci. 2022 Nov 22;15:972308. doi: 10.3389/fnmol.2022.972308.
- Huang B, Pan W, Wang W, Wang Y, Liu P, Geng W. Overexpression of Pleckstrin Homology Domain-Containing Family A Member 4 Is Correlated with Poor Prognostic Outcomes and Immune Infiltration in Lower-Grade Glioma. Dis Markers. 2022 Nov 11;2022:1292648. doi: 10.1155/2022/1292648.
- Li J, Wang J, Liu D, Tao C, Zhao J, Wang W. Establishment and validation of a novel prognostic model for lower-grade glioma based on senescence-related genes. Front Immunol. 2022 Oct 21;13:1018942. doi: 10.3389/fimmu.2022.1018942.
- Xu S, Wang Z, Ye J, Mei S, Zhang J. Identification of Iron Metabolism-Related Genes as Prognostic Indicators for Lower-Grade Glioma. Front Oncol. 2021 Sep 9;11:729103. doi: 10.3389/fonc.2021.729103.
- Li W, Ling L, Xiang L, Ding P, Yue W. Identification and validation of a risk model and molecular subtypes based on tryptophan metabolism-related genes to predict the clinical prognosis and tumor immune microenvironment in lower-grade glioma. Front Cell Neurosci. 2023 Feb 28;17:1146686. doi: 10.3389/fncel.2023.1146686.

Reviewer 2 Report
In this paper, authors provide transcriptomic analyses of Extracellular Matrix Protein 2 (ECM2) that allowed to associate its significant upregulation to high-grade Gliomas, and through immunohistochemical analyses in Human Protein Atlas it was confirmed that the level of ECM2 expression increases with the Glioma grade. Functional enrichment analyses were then performed to build the gene-network related to ECM2, which was found to be associated with different signaling pathways such as NFKB, inflammatory response, JAK-STAT3, IL-STAT5 and the complement system. Also, immune-related analyses showed a positive correlation between infiltrative immune cells and ECM2 levels, resulting in a worse prognosis compared to low-ECM2 and low-infiltrative cells conditions. IC50 values of different chemotherapy agents were tested, demonstrating that patients with high ECM2 levels are more sensitive to chemotherapy.
Interestingly, mutation frequencies of IDH1, CIC, FUBP1, NOTCH1, ARID1A, and IDH2 were significantly higher in the low ECM2 expression group, while the mutation frequencies of TP53, ATRX, EGFR, and PTEN were significantly higher in high ECM2 expression group. This results well correlate with the known literature, that associates IDH1 mutation with better prognosis, and EGFR and PTEN mutations with worse prognosis.
The authors propose ECM2 as a diagnostic marker since its expression is significantly upregulated in Gliomas and doesn’t show differences in different age levels and genders.
Furthermore, ECM2 knockdown led to a decrease in wound healing, migration and invasion ability of Glioma cells.
The work is well carried, and experiments gave clear results. I want to propose here few simple experiments that can improve this work:
1) The authors can look at protein expression of ECM2 should be performed and compared with transcriptomic analyses at least for the main experiment as sometimes mRNA levels does not necessarily predict the protein content amount.
2) Also, ECM2 role in migration and invasion should be further investigated: I suggest to better dissect the activated downstream pathways investigating the modulation of few intermediated that are modulated by ECM2 that could be targeted to inhibit tumor migration and invasion. At very least this point should be mentioned in the discussion section.
Author Response
Responds to the reviewer’s comments:
Reviewer #2:
General Comments: In this paper, authors provide transcriptomic analyses of Extracellular Matrix Protein 2 (ECM2) that allowed to associate its significant upregulation to high-grade Gliomas, and through immunohistochemical analyses in Human Protein Atlas it was confirmed that the level of ECM2 expression increases with the Glioma grade. Functional enrichment analyses were then performed to build the gene-network related to ECM2, which was found to be associated with different signaling pathways such as NFKB, inflammatory response, JAK-STAT3, IL-STAT5 and the complement system. Also, immune-related analyses showed a positive correlation between infiltrative immune cells and ECM2 levels, resulting in a worse prognosis compared to low-ECM2 and low-infiltrative cells conditions. IC50 values of different chemotherapy agents were tested, demonstrating that patients with high ECM2 levels are more sensitive to chemotherapy. Interestingly, mutation frequencies of IDH1, CIC, FUBP1, NOTCH1, ARID1A, and IDH2 were significantly higher in the low ECM2 expression group, while the mutation frequencies of TP53, ATRX, EGFR, and PTEN were significantly higher in high ECM2 expression group. This results well correlate with the known literature, that associates IDH1 mutation with better prognosis, and EGFR and PTEN mutations with worse prognosis. The authors propose ECM2 as a diagnostic marker since its expression is significantly upregulated in Gliomas and doesn’t show differences in different age levels and genders. Furthermore, ECM2 knockdown led to a decrease in wound healing, migration and invasion ability of Glioma cells. The work is well carried, and experiments gave clear results. I want to propose here few simple experiments that can improve this work.
Reply: Thank you for your insightful comment. We appreciate your professional opinion and expertise. Your comment reflects a deep understanding of the topic.
1. Comment: The authors can look at protein expression of ECM2 should be performed and compared with transcriptomic analyses at least for the main experiment as sometimes mRNA levels does not necessarily predict the protein content amount.
1. Reply: Thank you so much for your rigorous consideration. We compared the expression level of ECM2 and found that the expression of ECM2 significantly increased with the grade of glioma. We then explore the protein expression of ECM2 with immunohistochemical (IHC) staining and found that the results showed good consistency with the gene expression results.The results are shown below.
2. Comment: Also, ECM2 role in migration and invasion should be further investigated: I suggest to better dissect the activated downstream pathways investigating the modulation of few intermediated that are modulated by ECM2 that could be targeted to inhibit tumor migration and invasion. At very least this point should be mentioned in the discussion section.
2. Reply:Thank you very much for your valuable comment. We really appreciate your nice suggestion. However, due to the limited research on the ECM2 gene, the downstream pathway of ECM2 has not been discussed yet. Therefore, we predict the downstream pathway of ECM2 based on the results of GSEA analyses. The results havebeen shown below. We considered that ECM2 might promote tumor migration and invasion via the TNFα/NF-κB signaling, JAK/STAT3 signaling pathway, STAT5 signaling, KRAS signaling, P53 signaling, and PI3K/AKT/mTOR signaling pathways. We are going on the exploration of the signaling pathway of ECM2 in glioma by experiments in future studies.
We really appreciate your efforts on our paper and hope that the response will meet with approval. Thanks again for your careful review.

Reviewer 3 Report
This original study is very intresting, but results are poorly describe and it's diffucult to understand data correlation. Can authors implement the description of data presented?
The figure of wound healing assay is not clear because it's low contrasted. It's difficult to identify the cells inside the scratch as well as cell confluence. I suggest to improve the quality of this image.
Author Response
Responds to the reviewer’s comments:
Reviewer #3:
General Comments: This original study is very intresting, but results are poorly describe and it's diffucult to understand data correlation. Can authors implement the description of data presented?
Reply: Thank you very much for your valuable comment. We seriously considered your nice suggestion and revised our manuscript thoroughly by native speakers.
1. Comment: The figure of wound healing assay is not clear because it's low contrasted. It's difficult to identify the cells inside the scratch as well as cell confluence. I suggest to improve the quality of this image.
1. Reply: Thank you so much for your rigorous consideration. We have highlighted the cells inside the scratch as well as cell confluence to increase the contrast. We have refined the figure of wound healing assays and shown below.
Thanks for your careful review. We really appreciate your efforts on our paper and hope that the response will meet with approval. Your careful review has helped to make our study clearer and more comprehensive.

Reviewer 4 Report
In the manuscript 'High ECM2 Expression Predicts Poor Clinical Outcome and Promotes Proliferation, Migration and Invasiveness of Glioma', the authors have characterized the role of Extracellular Matrix Protein 2 (ECM2) in lower-grade gliomas (LGGs). The findings in the manuscript are interesting and the manuscript can be improved by addressing the following concerns:
1) The authors should extensively proofread the manuscript for typos and grammatical errors.
2) The rationale for investigating the role of ECM2 in LGG is unclear. Why did the authors decide to study ECM2 in particular and not ECM1?
3) Based on literature and the database analysis, can the authors speculate the mechanism of action of ECM2 in LGG?
4) ECM2 has a role in cell adhesion and extracellular matrix assembly from literature studies. It is expected to regulate cell migration/invasion and proliferation. Can the authors include a functional assay to demonstrate ECM2 as a unique prognostic marker in terms of immune infiltration?
5) One of the main drawbacks of this study is the lack of experimental validation in regard to immune infiltration. The findings from the database studies are interesting but should be validated experimentally to derive meaningful conclusions.
6) Figure 9: U87 and U251 are likely glioblastoma cell lines. Can they be used as models for LGG?
7) Have the authors compared the expression of ECM2 in U87 and U251 cells vs non-tumorigenic control?
8) Drug sensitivity assay (IC50 analysis) can be performed in ECM2 knockdown U87/U251 cells to experimentally validate whether knocking down ECM2 changes sensitivity towards the mentioned drugs in Section 3.4
Author Response
Responds to the reviewer’s comments:
Reviewer #4:
General Comments: In the manuscript 'High ECM2 Expression Predicts Poor Clinical Outcome and Promotes Proliferation, Migration and Invasiveness of Glioma', the authors have characterized the role of Extracellular Matrix Protein 2 (ECM2) in lower-grade gliomas (LGGs). The findings in the manuscript are interesting and the manuscript can be improved by addressing the following concerns.
Reply: Thank you very much for your valuable comment. We really appreciate your professional opinion and expertise. Your comment reflects a deep understanding of the topic.
1. Comment: The authors should extensively proofread the manuscript for typos and grammatical errors.
1. Reply: We are very sorry for the typos and grammatical errors in manuscript and the inconvenience they caused in your reading. We seriously considered your nice suggestion and revised our manuscript thoroughly by native speakers.
2. Comment: The rationale for investigating the role of ECM2 in LGG is unclear. Why did the authors decide to study ECM2 in particular and not ECM1?
2. Reply: Thank you very much for your rigorous consideration. We analyze the expression of ECM1 in different grades of glioma by TCGA database.We compare the survival distribution between the low ECM1 and high ECM1 groups and found that high ECM1 expression was significantly associated with the poor outcome in LGG patients.We further analyze the prognostic role of ECM1 in LGG by univariate and multivariate Cox regression analyses, however, the results are not significant. The results are shown below. Your nice suggestion provides a new perspective for us to further study the role of extracellular matrix in glioma.
3. Comment: Based on literature and the database analysis, can the authors speculate the mechanism of action of ECM2 in LGG?
3. Reply: Thank you very much for your valuable comment. We really appreciate your nice suggestion. However, due to the limited research on the ECM2 gene, the downstream pathway of ECM2 has not been discussed yet. Therefore, we predict the downstream pathway of ECM2 based on the results of GSEA analyses. The results havebeen shown below. We considered that ECM2 might promote tumor migration and invasion via the TNFα/NF-κBsignaling, JAK/STAT3 signaling pathway, STAT5 signaling, KRAS signaling, P53 signaling, and PI3K/AKT/mTOR signaling pathways. We are going on the exploration of the signaling pathway of ECM2 in glioma by experiments in future studies. In addition, it is noteworthy that ECM2 plays a potential role in immune infiltration and immune regulation, suggesting that ECM2 might be a potential target for immunotherapy.
4. Comment: ECM2 has a role in cell adhesion and extracellular matrix assembly from literature studies. It is expected to regulate cell migration/invasion and proliferation. Can the authors include a functional assay to demonstrate ECM2 as a unique prognostic marker in terms of immune infiltration?
4. Reply: Thank you very much for your valuable comment. We really appreciate your nice suggestion. In this study, we identified the potential role of ECM2 in immune regulation of glioma by GO and KEGG analyses. We further discussed the correlation between ECM2 and immune infiltration with two methods. There is a potential combined effect between ECM2 expression and immune infiltration in the prognosis of glioma patients as the Figure 3d showed. This is a preliminary study on the role of ECM2 in glioma, we did not conduct ECM2-related immune experiments. We will verify the role of ECM2 in immune infiltration in future studies.
5. Comment: One of the main drawbacks of this study is the lack of experimental validation in regard to immune infiltration. The findings from the database studies are interesting but should be validated experimentally to derive meaningful conclusions.
5. Reply: Thank you very much for your rigorous consideration. We totally agree with your opinion. The function of ECM2 in immune infiltration based on experimental verification is necessary. There were no previous studies on ECM2 in glioma and its role in immune infiltration.This study is a preliminary explorationand our following work will focus on the immune regulatory role of ECM2 in glioma. Thanks again for your professional opinion.
6. Comment: Figure 9: U87 and U251 are likely glioblastoma cell lines. Can they be used as models for LGG?
6. Reply: Thank you very much for your rigorous consideration. We totally understand your concern. The glioma cell lines U87 and U251 are stable cell lines, which are commonly used for cell experiments and functional verification of genes in glioma, including LGG. These two cell lines have been used in previous studies on LGG [1-5].These two cell lines were used to verify the role of ECM2 in glioma.
- Gu S, Shu L, Zhou L, Wang Y, Xue H, Jin L, Xia Z, Dai X, Gao P, Cheng H. Interfering with CALCRL expression inhibits glioma proliferation, promotes apoptosis, and predicts prognosis in low-grade gliomas. Ann Transl Med. 2022 Dec;10(23):1277. doi: 10.21037/atm-22-5154.
- Liu Z, Peng L, Sun Y, Lu Z, Wu B, Wang W, Zhang X, Hao H, Gong P. COMMD4 is a novel prognostic biomarker and relates to potential drug resistance mechanism in glioma. Front Pharmacol. 2022 Sep 30;13:974107. doi: 10.3389/fphar.2022.974107.
- He J, Li X, Zhu W, Yu Y, Gong J. Research of differential expression of sIL1RAP in low-grade gliomas between children and adults. Brain Tumor Pathol. 2018 Jan;35(1):19-28. doi: 10.1007/s10014-017-0304-x. Epub 2017 Dec 13.
- Tang L, Deng L, Bai HX, Sun J, Neale N, Wu J, Wang Y, Chang K, Huang RY, Zhang PJ, Li X, Xiao B, Cao Y, Tao Y, Yang L. Reduced expression of DNA repair genes and chemosensitivity in 1p19q codeleted lower-grade gliomas. J Neurooncol. 2018 Sep;139(3):563-571. doi: 10.1007/s11060-018-2915-4. Epub 2018 Jun 19.
- Zhang B, Xu C, Liu J, Yang J, Gao Q, Ye F. Nidogen-1 expression is associated with overall survival and temozolomide sensitivity in low-grade glioma patients. Aging (Albany NY). 2021 Mar 18;13(6):9085-9107. doi: 10.18632/aging.202789. Epub 2021 Mar 18.
7. Comment: Have the authors compared the expression of ECM2 in U87 and U251 cells vs non-tumorigenic control?
7. Reply: Thank you very much for your rigorous consideration. According to the results of RNA sequencing, the expression of ECM2 in glioma samples was significantly higher compared to normal tissues, including LGG and GBM. The aim of cell experiments was to explore and confirm the potential function of ECM2 in gliomas. According to the previous studies [6-10], no comparison was performed between glioma cell lines and non-tumorigenic controls.
- Han W, Chen L. Predictive significance of PRR11 in prognosis and immune infiltration of glioma patients. Mol Carcinog. 2023 Apr 10. doi: 10.1002/mc.23539.
- Wang Y, Zhao D, Wang H, Wang S, Zhang H, Liu H, Wang K. Long non-coding RNA-LINC00941 promotes the proliferation and invasiveness of glioma cells. Neurosci Lett. 2023 Jan 31;795:136964. doi: 10.1016/j.neulet.2022.136964.
- Xiong Z, Wu S, Li FJ, Luo C, Jin QY, Connolly ID, Hayden Gephart M, You L. Elevated ETV6 Expression in Glioma Promotes an Aggressive In Vitro Phenotype Associated with Shorter Patient Survival. Genes (Basel). 2022 Oct 17;13(10):1882. doi: 10.3390/genes13101882.
- Dubois N, Berendsen S, Tan K, Schoysmans L, Spliet W, Seute T, Bours V, Robe PA. STAT5b is a marker of poor prognosis, rather than a therapeutic target in glioblastomas. Int J Oncol. 2022 Oct;61(4):124. doi: 10.3892/ijo.2022.5414.
- Sun W, Zou Y, Cai Z, Huang J, Hong X, Liang Q, Jin W. Overexpression of NNMT in Glioma Aggravates Tumor Cell Progression: An Emerging Therapeutic Target. Cancers (Basel). 2022 Jul 21;14(14):3538. doi: 10.3390/cancers14143538.
8. Comment: Drug sensitivity assay (IC50 analysis) can be performed in ECM2 knockdown U87/U251 cells to experimentally validate whether knocking down ECM2 changes sensitivity towards the mentioned drugs in Section 3.4.
8. Reply: Thank you very much for your rigorous consideration. We totally agree with your opinion. Drug susceptibility assay has important guiding significance for the individual treatment. This study is a preliminary exploration, we did not conduct drug susceptibility assay in this study. We will verify the drug susceptibility assay in future studies.
Thanks for your careful review. We really appreciate your efforts on our paper and hope that the response will meet with approval. Your careful review has helped to make our study clearer and more comprehensive.

Round 2
Reviewer 1 Report
The authors clarified a few points. However, I still think the nomenclature (arabic numerals) of the current WHO classification should be applied.
Furthermore, the authors should include a section discussion regarding the lack of data regarding extent of resection and completion of adjuvant radiochemotherapy. If you take a look at the significant higher proportion of IDH-1 wild-type gliomas, there is a high probability that those patients are older, have a worse KPS, underwent biopsy only, or do not complete radiochemotherapy.
Those real-life variables which cannot be included in this analysis should be at least discussed. Furthermore, I would agree that high ECM2 expression is associated with survival, but it is not a predictor.
Lines 67-70 describe results. This should be removed from an introduction.
Author Response
Responds to the reviewer’s comments:
Reviewer #1:
1. Comment:
The authors clarified a few points. However, I still think the nomenclature (arabic numerals) of the current WHO classification should be applied.
1. Reply:
Thanks for your suggestions. We totally agree with your opinion. Applied current WHO classification could be more convincible. However, this was a retrospective study. The RNA-seq and the matching clinical data for analyses were obtained from TCGA database (https://portal.gdc.cancer.gov). The gene expression data and clinical features for validation were obtained from CGGA database (http://www.cgga.org.cn/), REMBRANDT cohort (http://www.betastasis.com/glioma/rembrandt/), and GSE16011 dataset (https://www.ncbi.nlm.nih.gov/gds/). The information is widely used in recent articles. Sorry for the misunderstandings. Thanks again for your professional opinion.
2. Comment:
Furthermore, the authors should include a section discussion regarding the lack of data regarding extent of resection and completion of adjuvant radiochemotherapy. If you take a look at the significant higher proportion of IDH-1 wild-type gliomas, there is a high probability that those patients are older, have a worse KPS, underwent biopsy only, or do not complete radiochemotherapy.
Those real-life variables which cannot be included in this analysis should be at least discussed. Furthermore, I would agree that high ECM2 expression is associated with survival, but it is not a predictor.
2. Reply:
Thanks for your suggestions. We have added the following content in the section of Discussion. “Clinical predictors are important for overall survival of gliomas, such as extent of resection, radiotherapy and chemotherapy. However, the information was quite incomplete in TCGA database. Therefore, we further included the information of radiotherapy and chemotherapy data in CGGA mRNAseq-693 dataset and CGGA mRNAseq-325 dataset to explore the prognostic role of ECM2. The results of Cox regression showed that ECM2 was still independently associated with the prognosis of LGG patients, even after the inclusion of radiotherapy and chemotherapy (P<0.05 for both). Clinical indicators are also needed in future studies.”
Our study revealed that high ECM2 expression was significantly associated with the poor outcome on univariate and multivariate Cox analyses (P<0.05). After knockdown of ECM2, the functional experiments showed a significant decrease in proliferation, migration, and invasion in glioma cell lines. The ECM2 do have prognostic value in LGG patients. Thanks again.
3. Comment:
Lines 67-70 describe results. This should be removed from an introduction.
3. Reply:
Thanks for your suggestions. We have removed the corresponding content.
Thanks for your careful review. We really appreciate your efforts on our paper and hope that the response will meet with approval. Your careful review has helped to make our study clearer and more comprehensive.

Reviewer 2 Report
No more request from my side.
I support the publication of the following manuscript.
Kind regards
Author Response
Thanks for your careful review. We really appreciate your efforts on our paper.
Reviewer 3 Report
The authors responded to the previous requests, also improving the quality of the wound healing assay image
Author Response

(The authors gave the same response as above.)

Reviewer 4 Report
All comments have been addressed and the manuscript is suitable for publication
Author Response

(The authors gave the same response as above.)
